# Subjective Wellbeing and Its Associated Factors among University Community during the COVID-19 Pandemic in Northern Malaysia

**DOI:** 10.3390/healthcare10061083

**Published:** 2022-06-10

**Authors:** Syaheedatul Iman Dinsuhaimi, Asrenee Ab Razak, Ahmad Tajudin Liza-Sharmini, Wan Mohd Zahiruddin Wan Mohammad, Azhany Yaakub, Azizah Othman, Aziah Daud, Kamarul Imran Musa, Nani Draman, Alwi Besari

**Affiliations:** 1Department of Psychiatry, School of Medical Sciences, Universiti Sains Malaysia, Kubang Kerian 16150, Malaysia; drimah@usm.my; 2Department of Ophthalmology, School of Medical Sciences, Universiti Sains Malaysia, Kubang Kerian 16150, Malaysia; liza@usm.my (A.T.L.-S.); azhany@usm.my (A.Y.); 3Department of Community Medicine, School of Medical Sciences, Universiti Sains Malaysia, Kubang Kerian 16150, Malaysia; drzahir@usm.my (W.M.Z.W.M.); aziahkb@usm.my (A.D.); drkamarul@usm.my (K.I.M.); 4Department of Paediatrics, School of Medical Sciences, Universiti Sains Malaysia, Kubang Kerian 16150, Malaysia; azeezah@usm.my; 5Department of Family Medicine, School of Medical Sciences, Universiti Sains Malaysia, Kubang Kerian 16150, Malaysia; drnani@usm.my; 6Department of Internal Medicine, School of Medical Sciences, Universiti Sains Malaysia, Kubang Kerian 16150, Malaysia; dralwi@usm.my

**Keywords:** subjective wellbeing, university community, lockdown, COVID-19

## Abstract

Lockdown implementation during COVID-19 pandemic has caused many negative impacts in various aspect of life, including in the academic world. Routine disruption to teaching and learning environment has raised concerns to the wellbeing of university staff and students. This study aimed to examine the subjective wellbeing of the university community in Northern Malaysia during lockdown due to COVID-19 pandemic and the factors affecting it. An online cross-sectional survey involving 1148 university staff and students was conducted between March and April 2020. The research tools include the Personal Wellbeing Index (PWI) to assess subjective wellbeing and the Depression, Anxiety and Stress 21 (DASS-21) scale for psychological distress. While we found the subjective wellbeing score in our study population was stable at 7.67 (1.38), there was high prevalence of anxiety, depression, and stress with 27.4%, 18.4%, and 11.5%, respectively. The students reported higher levels of psychological distress compared to staff. The PWI score was seen to be inversely affected by the depression and stress score with a reduction in the PWI score by 0.022 (95% CI −0.037 to −0.007) and 0.046 (95% CI −0.062 to −0.030) with every one-unit increment for each subscale, respectively. Those who perceived to have more difficulty due to the lockdown also reported low subjective wellbeing. Thus, it is crucial to ensure policies and preventative measures are in place to provide conducive teaching and learning environment. Additionally, the detrimental psychological effects especially among students should be addressed proactively.

## 1. Introduction

In December 2019, coronavirus disease 2019, otherwise known as COVID-19, was discovered in Wuhan, China. By the end of March 2020, the disease had spread to 171 countries, with a mortality rate of 4.5% worldwide [1]. In February, the first verified case was reported in Malaysia, and from then on, cases grew in number, prompting the country’s full lockdown during the first wave of the pandemic. During the lockdown, only a few essential sectors were allowed to function while other services were shut down, including the education sector, and the public were not allowed to go out of their house without necessity [1]. The large number of cases and deaths has triggered many negative reactions in people, including fear of becoming infected with the disease and the infection of loved ones [2,3,4]. Reading negative news regularly, receiving ambiguous information, or believing in ambiguous messages about COVID-19 can all have detrimental influences on psychological wellbeing, which may be intensified by the unpleasant feelings experienced from the COVID-19 pandemic [5]. This would lead to effects on one’s psychological status, such as depression, anxiety, stress, and trauma. A study in China during the early period of the COVID-19 pandemic found that 28.8%, 16.5%, and 8.1% of 1210 respondents had moderate to severe depression, anxiety, and stress, respectively [6]. The study also found that psychological distress was associated with being female, a student, and having physical symptoms.

The lockdown had further exacerbated adverse psychosocial implications in the public amid this anxiety-inducing pandemic. The university community, one of the nations’ fundamental resources of education, was not spared from these negative impacts. University students and staff have been reported to have higher stress due to disruption of academic years, the postponement of examinations, needing to adapt to online work or classes, and social isolation, and all this in addition to the fear of COVID-19 [7,8,9,10]. A study among 2530 members of a university in Spain during the initial period of lockdown revealed that 21.34%, 34.19%, and 28.14% had moderate to extremely severe scores for anxiety, depression, and stress, respectively [11]. In Malaysia, 2.8% was observed to have moderate to severe anxiety in a study among university students during this pandemic [12].

These negative psychological impacts from COVID-19 could affect the nation’s subjective wellbeing (SWB). Subjective wellbeing is an overall appraisal of one’s life across different aspects, which involves satisfaction with life as a whole or in specific domains and affects [13]. This perception of satisfaction with life is strongly influenced by emotional experiences. While positive emotions tend to be associated with higher SWB, negative emotions affect the opposite [14,15]. The importance of SWB should be noted as it is correlated to mental health and not as a continuum end. Although SWB can overlap with psychological distress, the former gives a more holistic view of a person’s quality of life. The presence of psychopathology does not necessarily suggest absence of mental health and good wellbeing, and vice versa. Previous findings found high levels of subjective wellbeing in patients with mental illness despite suffering from high levels of psychopathology [16]. Wellbeing or positive mental health is a valuable component that may protect against the negative mental health impacts during stressful events. High SWB has been linked to better productivity and achievements, good mental and physical health, and positive relationships [14,15]. A study in 2018 found that good wellbeing can moderate the association between depression and suicidal ideation and act as a protective factor [16]. Previous data have shown that the wellbeing score is lower during this pandemic due to the increase in negative effects psychologically [5,17]. However, these studies used instrument that measures the subjective wellbeing broadly, and satisfaction in specific domains were not analyzed.

In a longitudinal sample of Australian adults showed that the pandemic has negatively affected the subjective wellbeing with a drop during the lockdown period [18]. While in Spain and New Zealand, the wellbeing remained stable [5,18]. Some of the factors seen to be affecting the subjective wellbeing was unemployment or loss of income due to COVID-19, younger age [19], higher perceived stress [20], and fear of COVID-19 infection [2,3]. A study among 80 surgeons in Canada found an association between depression and emotional distress with lower satisfaction in personal safety, future security and personal achievement [21].

During the period of this study, there were limited local studies examining the personal wellbeing of the university community. Being a factory of academicians and professionals, the university is one of the nation’s most precious assets; as such, it is critical to monitor its well-being so that required policies and support can be instated. Thus, this study’s goal was to assess the psychological status and to determine the level of subjective wellbeing of the university community during the first lockdown and factors associated with it.

## 2. Materials and Methods

This cross-sectional study was conducted from March until May 2020, during the full lockdown of the pandemic’s first wave in Malaysia. A set of online self-administered questionnaires were distributed among the staff and students at a public university in Northern Malaysia. This university has three campuses, which the main campus located in northwest, the health campus in the northeast, and the engineering campus in the northwest. This university has approximately 30,000 staff, which comprises administrative, academicians, and supporting workers, and 6000 undergraduate and postgraduate students. In the health campus, most of the staff and students are involved with healthcare. The questionnaires were distributed through the university’s emailing system and social media application (WhatsApp and Telegram) via group representatives. This study used a convenience sampling method. The respondent must be literate in Malay language and registered as a staff or student at the university during the period of sampling to participate in the study. They have also given consent prior to submitting responses. A total of 1220 responded to the questionnaires; however, 72 of them were excluded due to incompletion. The ratio of staff to student who responded to this study was around 4:1. Ethical clearance was given by The Human Research Ethics Committee of the university.

The online questionnaire was transformed into Google Form, consisted of pre-structured sociodemographic proforma designed for the study. Depression, Anxiety and Stress Scale-21 (DASS-21) and Personal Well-Being Index (PWI) were used to assess the psychological distress and personal wellbeing score, respectively. In the sociodemographic proforma, we included age, gender, ethnic groups, religion, marital status, level of education, which campus, occupation (student or staff), healthcare worker (frontliner and non-frontliner), head of family, living arrangements, and any history of personal psychiatry illness or in the family. In addition, other factors related to the pandemic and lockdown, such as exposure to COVID-19, whether they have been infected with COVID-19 or had been in close contact with an infected person, person under investigation (PUI) for having symptoms, treating team of COVID-19, or volunteering at a COVID-19 center; were included. Participants were asked whether they received any aids during the lockdown, such as food, government financial incentives, and others. The questionnaire also included receiving support from others including family, friends, social media, or superiors. Perceived level of difficulty during the lockdown was also collected.

### 2.1. Instruments

#### 2.1.1. Depression, Anxiety and Stress Scale-21 (DASS-21)

The levels of psychological disturbances were measured using the validated Malay version of Depression, Anxiety and Stress Scale-21 (DASS-21). This self-report questionnaire has 21 items which are grouped into three subscales of depression, anxiety, and stress. It has a good validity, with Cronbach’s alpha values of 0.84, 0.74, and 0.79 for depression, anxiety, and stress, respectively [22]. Respondents would rate items using a 4-point severity scoring of 0–3, and the sum of each subscale are then multiplied by two, giving a score range of 0–42 for each subscale. In this study, the cut-off scores of 9, 7, and 14 were used for depression, anxiety, and stress, respectively, based on the scoring system proposed. A higher the score for each subscale depicts higher distress, and it can be further categorized into mild, moderate, severe, and extremely severe based on the suggested cut-off points in a previous study [23].

#### 2.1.2. Personal Well-Being Index (PWI))

The Personal Well-being Index (PWI) was developed by the International Wellbeing Group (IWbG) and has been used in many countries to measure the community’s subjective well-being [24]. They have translated the index into multiple language including Malay version. In Malaysia, the Malay version of PWI has been integrated into the Fifth Malaysian Population and Family Survey (MPFS-5) by National Population and Family Development Board, Malaysia [25]. The PWI instrument measures satisfaction level in 8 domains, which include standard of living, personal health, achieving in life, personal relationships, personal safety, community-connectedness, security and religiosity. Respondents are required to rate each domain with an 11-point scale, ranging from a scale of 0, meaning “dissatisfied” (completely dissatisfied), to 10 points, meaning “fully satisfied” (completely satisfied). The average score of all domains will represent a person’s subjective well-being at a maximum of 10 [26]. Cronbach’s alpha for the PWI Malay version in this study was 0.93, higher than that found in Australia and overseas studies, which lies between 0.70 and 0.85 [26].

### 2.2. Data Analyses

All data were analyzed using the IBM Statistical Package for the Social Sciences (SPSS) Version 26 (IBM, Armonk, NY, USA). The socio-demographics of the participants, the psychological status of depression, anxiety, and stress and the personal wellbeing index score were presented using descriptive analysis. Categorical variables were described as frequency and percentage of answers, while continuous variables were presented as mean and standard deviation (SD). Regression analysis was carried out to determine the significant associations with subjective wellbeing score (derived from PWI). Variables from sociodemographic characteristics, factors related to COVID-19 and lockdown, and DASS-21 scores were initially entered in the simple linear regression to assess individual association with subjective wellbeing score. Factors with *p*-value of less than 0.25 or clinically significant were selected and entered into the multiple linear regression. Stepwise, forward and backward method were applied. For the final model, factors with *p*-value < 0.05 were chosen as statistically significant. R square of the final model is checked for model fitness.

## 3. Results

### 3.1. Sociodemographic Characteristics

Table 1 shows that majority of the participants were female (76.4%), married (72.1%), Malay (91.4%), Muslim (92.2%), and certificate or diploma holder (56.5%). Respondents were mainly from the Health Campus (83%), and 833 (72.6%) were healthcare workers. Of the 246 students who participated, 17% were full time students. Most of the respondents lived in their own house (49.7%) during the lockdown.

Relating to the COVID-19 pandemic and lockdown, a small portion received essential aids during the lockdown, and more than half (64.6%) answered that they have received moral support either from family, friends, the workplace, superiors, or social media. Among the respondents, only 202 (17.7%) had exposure to COVID-19. Meanwhile, almost half reported having moderate to extreme difficulty due to lockdown.

### 3.2. Psychological Status and Personal Wellbeing

Among 1148 respondents, around 27.4% were found to have moderate to very severe anxiety. While 18.4% and 11.5% of them had moderate to very severe depression and stress, respectively. Table 2 and Table 3 provide a breakdown of level of depression, anxiety, and stress according to the DASS-21 among staff and students, respectively. Higher levels of psychological distress were found in the students’ population compared to staff.

The subjective wellbeing had a mean of 7.67 (1.38). The highest reported satisfaction was in the personal health domain, with a mean of 7.98 (1.68). Three domains had relatively low scores, which were future security, achieving in life, and community-connectedness, with a range of 7.44–7.50 and compared to other domains with score of 7.55 and more. Other domains’ mean scores are as shown in Table 4.

### 3.3. Factors Associated with Personal Wellbeing

The linear regression analysis in Table 5 shows that subjective wellbeing, which is reflected by the Personal Wellbeing Index (PWI), was significantly associated with age, marriage status, being a student, perceived level of difficulty due to lockdown, depression, and stress. Higher PWI scores were found to be associated with the increment in age (b = 0.016, 95% CI 0.008 to 0.025). Being married was associated with a higher PWI score by 0.189 (95% CI 0.018–0.359) compared to single or divorced individuals. Lower PWI score was associated with being a student compared to staff working in university (b = −0.206, 95% CI −0.390 to 0.022). In relation to the lockdown, those who perceived to have moderate to extreme difficulty due to lockdown would score lower in the PWI (b = −0.363, 95% CI −0.496 to −0.231) compared to those who did not have any or little difficulty. In terms of psychological status, the PWI score was seen to be inversely affected by the depression and stress score with a reduction in the PWI score by 0.022 (95% CI −0.037 to −0.007) and 0.046 (95% CI −0.062 to −0.030) with every one-unit increment for each subscale, respectively.

## 4. Discussion

This study examined the prevalence of psychological status, and level of personal wellbeing and its associated factor among university staff and students in Malaysia during the first lockdown. The respondents were mostly female, married, and had good educational backgrounds. Furthermore, the majority of them are from the health campus and are involved as healthcare workers.

The prevalence of psychological distress found in this study is comparable to the prevalence in the general population across the world during the COVID-19 pandemic [27]. However, compared to a study among a university community in Spain, the numbers are lower. The explainable reasons of the difference in findings could be that despite having a similar sample population, the previous study had more participants among students (76.8%) compared to this study (21.4%) [11]. Nonetheless, the high prevalence of psychological distress among students in this study is in line with previous studies [7,11]. Students are known to have higher levels of stress, even in the pre-pandemic era, which might explain the differences in the findings [28]. Apart from that, the situation of the pandemic during which the study was conducted might also play a role. Although going through lockdown, the pandemic situation in Malaysia was fairly controlled, with a low number of cases and deaths at the time of the study, whereas Spain had the third-largest number of cases of COVID-19 and deaths [11]. This is reflected in the study, where only 17.7% of the respondents were exposed to COVID-19. The effect of psychological distress might not be seen as significant yet compared to those dealing with COVID-19. Thus, it is crucial to note that psychological distress might worsen if the pandemic is not properly controlled [29].

Subjective wellbeing is a broader definition of mental health than simply being mentally unwell. It is comprised of life satisfaction and affections. Wellbeing is correlated to positive affection and inversely influenced by negative affections [29]. The theoretical range of life satisfaction of a group population was deduced to be around the mean of 70% or 7.00, with a narrow positive range of values. This is thought to be due to the internal homeostatic control of life satisfaction despite having external stressors. However, values lower than that would mean that the population is experiencing homeostatic failure [30]. Thus, in this study, we would not expect the Personal Wellbeing Index (PWI) score to be greatly impacted with the current situation of the pandemic in Malaysia. Our finding replicated the wellbeing of New Zealand’s population during the lockdown, compared to that seen in Australia, where the wellbeing score dropped 10 marks from pre-lockdown. This is most probably because of the rate of COVID-19 cases in Malaysia and New Zealand were lower compared to Australia [18]. The subjective wellbeing score in times of lockdown among the university members was found to be slightly lower than the general population in Malaysia before lockdown, which was 7.71 in 2014 [25]. Compared to a study among women during the COVID-19 pandemic, the mean range of the PWI across the domains were higher compared to the PWI of 7.21 to 7.71 (SD: 1.79–1.92) in this study [31]. It is worth noting that this study mostly studied those who have higher socioeconomic backgrounds compared to the previous ones. Interestingly, we found that satisfaction in personal health had the highest mean in the time of this health-worrying pandemic, and it was even higher compared to the pre-pandemic period. Feeling gratitude for having a lower rate of exposure to COVID-19 among participants might have a positive influence on their subjective health [32]. However, on the other hand, fear of becoming infected themselves or the infection of family and the ambiguity of the progression of the lockdown and pandemic might explain the lower satisfaction of their future security [2].

Subjective wellbeing among the university community was seen to be higher with increasing of age and being married compared to younger age and other marital status counterpart. Wellbeing score has been observed to be reduced with older age in previous studies, which contradicted to our findings [15,33]. However, a survey among women aged below 59 years old in Malaysia showed the replication of our study’s findings. In our samples, older participants were mostly married and had stable employment, which may suggest that they are provided with a more stable life and social support during the lockdown and thus higher satisfaction with life [34]. Previously, people with a higher satisfaction with life were found to be more likely to have stronger social connections than those with a lower satisfaction, which is reflected by this study’s findings [13,35].

Depression and stress were found to have significant negative relationships with subjective wellbeing, which is consistent with earlier findings. Having a good subjective wellbeing is protective towards depressive symptoms and other psychological distress [13]. Individuals who are depressed have been shown to have a pervasive and general pessimistic attitude, with a proclivity to ascribe unpleasant occurrences (more so over during this pandemic) to stable, internal, and global elements, which may impact their evaluations of personal wellbeing [36]. This study found lower subjective wellbeing in university students compared to staff during the time of lockdown. This is correlated with the higher prevalence of psychological distress among students. During the lockdown, some students had to be socially isolated, which may cause lack of social support. A shift to online learning might also cause stress, especially to those with poor internet connection and lack of proper technology tools [3,12]. Furthermore, the postponement of semesters and examinations caused uncertainty in their future path, as well as their financial planning [8]. This is also consistent with the current finding that people who reported more difficulties because of the lockdown had poorer subjective wellbeing.

This study was among the earliest study at the time of the data collection to highlight the subjective wellbeing in a university environment in Malaysia during the COVID-19 pandemic and the factors that contributed to it, as well as the psychological status. Thus, it provides valuable insights to implement policies and psychological interventions necessary in universities during pandemics and a baseline data for future research. Regarding the limitation, the tool used (DASS-21) is able to screen for depression, anxiety, and stress; however, clinical diagnosis could not be made. Thus, future study could include a diagnostic tool for depression and anxiety to ascertain the prevalence of mental illness during this pandemic. Our findings could not be generalized due to the method of convenience sampling, which may limit the sample to represent demographic of the general population. Furthermore, the online self-reporting questionnaire invites response bias to the study and leaves out those without access to the internet. The sample population was taken from a single institution; thus, we might not have included those from low socioeconomic backgrounds. A larger sample from different universities should be studied to be able to generalize the findings. Associations for specific domains of the personal wellbeing were not studied, which may be useful in giving insight on specific unmet needs. In addition, a prospective study of these participants to see changes in subjective wellbeing scores in the progress of the pandemic would be enlightening; however, due to confidentiality, we were unable to retain contact details of the respondents for follow-up.

## 5. Conclusions

The level of psychological distress is high with slightly lower SWBs among university staff and students in Malaysia during the COVID-19 pandemic compared to the pre-pandemic period. Students, enduring difficulty during lockdown and higher depressive and stress symptoms, experienced negative effects to their SWB.

This study’s findings are a first step to monitor the university’s wellbeing; to ensure policies and preventative measures are in place; to provide a conducive environment and support for the community, especially to the students. Additionally, mental health care should be aimed at and emphasized on helping to reduce the psychological distress among the university community and to maintain good wellbeing during this crisis.

## Figures and Tables

**Table 1 healthcare-10-01083-t001:** Characteristics of participants (*n* = 1148).

Demographic Characteristics	Frequency (%)
Sex Male Female	271 (23.6) 877 (76.4)
Age	34.57 (8.88) *
Marital status Single Married Divorced	282 (24.6) 829 (72.2) 37 (3.2)
Ethnicity Malay Chinese Indian Others	1049 (91.4) 63 (5.4) 26 (2.3) 10 (0.9)
Religion Islam Buddha Christian Hindu Others	1058 (92.2) 53 (4.6) 19 (1.6) 15 (1.3) 3 (0.3)
Education level Secondary school Certificate/Diploma Degree Masters PhD	195 (17.0) 649 (56.5) 190 (16.6) 56 (4.9) 58 (5.0)
Campus Main campus Health Engineering	174 (15.2) 953 (83.0) 21 (1.8)
Student Staff	246 (21.4) 902 (78.6)
Current education intake (*n* = 246) Undergraduate Postgraduate	162 (14.1) 84 (7.3)
Healthcare workers No Yes	315 (27.4) 833 (72.6)
Frontliner No Yes	850 (74.0) 298 (26.0)
History of psychiatry illness No Yes	1127 (98.2) 21 (1.8)
Family history of psychiatry illness No Yes	1094 (95.3) 54 (4.7)
Stay during lockdown Own house Family/relative’s house Rented house/room Student accommodation/quarters	571 (49.7) 273 (23.8) 236 (20.6) 68 (5.9)
Exposure to COVID-19 No Yes	941 (82.3) 202 (17.7)
Positive COVID-19 No Yes	1132 (98.6) 16 (1.4)
Loss of family members to COVID-19 No Yes	1147 (99.9) 1 (0.1)
Receiving COVID-19 aid No Yes	979 (85.3) 169 (14.7)
Receiving moral support during lockdown No Yes	406 (35.4) 742 (64.6)

* Mean (SD).

**Table 2 healthcare-10-01083-t002:** Level of depression, anxiety, and stress (DASS-21) among university staff (*n* = 902).

Psychological Status	Depression	Anxiety	Stress
		Frequency (%)	
Normal	679 (75.3)	618 (68.5)	755 (83.7)
Mild	85 (9.4)	60 (6.7)	65 (7.2)
Moderate	85 (9.4)	131 (14.5)	42 (4.7)
Severe	25 (2.8)	42 (4.7)	29 (3.2)
Very severe	28 (3.1)	51 (5.7)	11 (1.2)

**Table 3 healthcare-10-01083-t003:** Level of depression, anxiety, and stress (DASS-21) among university students (*n* = 246).

Psychological Status	Depression	Anxiety	Stress
		Frequency (%)	
Normal	145 (58.9)	145 (58.9)	179 (72.8)
Mild	28 (11.4)	11 (4.5)	17 (6.9)
Moderate	38 (15.4)	38 (15.4)	22 (8.9)
Severe	13 (5.3)	14 (5.7)	19 (7.7)
Very severe	22 (8.9)	38 (15.4)	9 (3.7)

**Table 4 healthcare-10-01083-t004:** Personal Wellbeing Index score among university community (*n* = 1148).

Domains	Level of Satisfaction
	Mean (SD)
Standard of living	7.75 (1.71)
Personal health	7.98 (1.68)
Achieving in Life	7.49 (1.70)
Personal Relationships	7.74 (1.64)
Personal Safety	7.55 (1.71)
Community-Connectedness	7.50 (1.63)
Future Security	7.44 (1.75)
Spirituality or Religion	7.86 (1.65)
Subjective wellbeing (mean of 8 domains)	7.67 (1.38)

**Table 5 healthcare-10-01083-t005:** Factors associated with subjective wellbeing.

Variables	Simple Linear Regression		Multiple Linear Regression		
	Crude b (CI 95%)	*p*-Value	Adjusted b (CI 95%)	t-Stat	*p*-Value
Gender					
Male	1				
Female	0.080	0.387			
	(−0.101, 0.261)				
Age (years)	0.035	<0.001 *	0.016	3.861	<0.001 **
	(0.026, 0.043)		(0.008, 0.025)		
Status					
Single/divorced	1				
Married	0.650	<0.001 *	0.189	2.167	0.030 **
	(0.483, 0.818)		(0.018, 0.359)		
Ethnicity					
Malay	1				
Non-Malay	−0.258	0.064 *			
	(−0.532, 0.015)				
Education level					
Secondary school	1				
Certificate/Diploma	0.191	0.078 *			
	(−0.021, 0.402)				
Degree	0.047	0.725			
	(−0.217, 0.312)				
Masters/PhD	0.493	0.002 *			
	(0.187, 0.799)				
Campus					
Main campus/Engineering	1				
Health	0.183	0.080 *			
	(−0.022, 0.387)				
Student/Staff					
Student	1		1		
Staff	−0.722	<0.001 *	−0.206	−2.199	0.028 **
	(−0.904, −0.539)		(−0.390, −0.022)		
History of psychiatry illness					
No	1				
Yes	−1.617	<0.001 *			
	(−2.184, −1.051)				
Family history of psychiatry					
No	1				
Yes	−0.232	0.211 *			
	(−0.595, 0.131)				
Head of family					
No	1				
Yes	0.202	0.024 *			
	(0.027, 0.377)				
Living arrangements					
Own house	1				
Family’s house	−0.398	<0.001 *			
	(−0.585, −0.211)				
Rented house/room	−0.423	<0.001 *			
	(−0.620, −0.225)				
Student accommodation/quarters	−1.066	<0.001 *			
	(−1.393, −0.739)				
Healthcare worker					
No	1				
Yes	0.443	<0.001 *			
	(0.272, 0.613)				
Frontliner					
No	1				
Yes	0.176	0.034 *			
	(0.013, 0.339)				
Perceived level of difficulty due to lockdown					
No/little	1		1		
Moderate to extremely	−0.522	<0.001 *	−0.363	−5.384	<0.001 **
	(−0.673, −0.371)		(−0.496, −0.231)		
Exposure to COVID-19					
No	1				
Yes	−0.208	0.040 *			
	(−0.407, −0.010)				
Receive aid					
No	1				
Yes	−0.332	0.003 *			
	(−0.549, −0.116)				
Receive moral support					
No	1				
Yes	0.036	0.661			
	(−0.125, 0.197)				
DASS-21					
Depression score	−0.075	<0.001 *	−0.046	−5.717	<0.001 **
	(−0.083, −0.067)		(−0.062, −0.030)		
Anxiety score	−0.069	<0.001 *			
	(−0.078, −0.60)				
Stress score	−0.069	<0.001 *	−0.022	−2.864	0.004 **
	(−0.077, −0.061)		(−0.037, −0.007)		

* Significant variables with *p* < 0.25 were included in the multiple linear regression analysis ** Variables with *p* < 0.05 were retained for the final model. R^2^ = 28.8%. Stepwise, backward and forward multiple linear regression method applied. Model assumptions are fulfilled. There were no interactions amongst independent variables. No multicollinearity detected.

## Data Availability

The data presented in this study are available on request from the corresponding author. The data are not publicly available due to privacy.

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
