# Peer review of "Subjective Wellbeing and Its Associated Factors among University Community during the COVID-19 Pandemic in Northern Malaysia"

_healthcare, 2022, doi:10.3390/healthcare10061083_

Round 1

Reviewer 1 Report

The COVID-19 pandemic and lockdown implementation have caused many negative impacts to the nations around the world including to the university community. This could also affect the wellbeing of the members of the university

This authors aimed to examine the subjective wellbeing of university community in Malaysia during lockdown due to COVID-19 pandemic and factors affecting it. An online cross-sectional survey among the university staffs and students was conducted between March and April 2020. Questionnaires of sociodemographic data, Depression, Anxiety and Stress 21 (DASS-21) and Personal Wellbeing Index (PWI) were distributed. Among 1148 respondents, 27.4% found to have moderate to very severe anxiety, while 18.4% and 11.5% for depression and stress respectively.

They found that the  subjective wellbeing score of the university community was found to be stable at 7.67 (1.38), with highest satisfaction in the personal health domain and relatively lower in future security and achieving in life.

Very interesting they showed that: (a) higher wellbeing was associated with older age, being married and a student. (b) Those who perceived to have more difficulty due to the lockdown had lower PWI, as well as those with higher score of depression and stress.

They concluded that  although the pandemic has not seen to take a toll on the wellbeing, it is crucial to take initiative steps in maintaining the university community’s wellbeing by addressing certain aspects that could have detrimental effect by the pandemic in the long term.

They article is very interesting.

I suggest the following with a pure academic spirit:

  1. Insert a more clear and explicit purpose
  2. Insert a table with acronyms
  3. Results are arranged into themes. Introduce them by means of a few rows
  4. I suggest to rearrange table 1. It seems hard to follow

Author Response

Thank you for your review. We have addressed your comments accordingly as in the attached file.

Reviewer 2 Report

See uploaded document

Author Response

(The authors gave the same response as above.)

Round 2

Reviewer 2 Report

The language requires further editing still.

Well done to the authors for making these changes.